# Leveraging Classical and Quantum Computing for Process Systems Engineering Applications

Yirang Park[1][0009−0008−6629−3308] and David E. Bernal Neira[1][0000−0002−8308−5016]

Davidson School of Chemical Engineering, Purdue University, 480 Stadium Mall Drive, West Lafayette, IN 47907, USA
dbernaln@purdue.edu

**Abstract.** Mixed-integer nonlinear programming (MINLP) is a modeling paradigm that combines discrete and continuous variables to model and solve a wide range of optimization problems. Its flexibility is especially useful for many real-world decision problems in engineering, operations, and finance, as these problems often involve discrete decisions and nonlinear system behaviors [24]. Despite the ease of modeling, MINLP problems are challenging to solve as monolithic problems due to the combinatorial complexity of discrete variables and nonlinearities; however, they can be made manageable by adopting a decomposition strategy. Additionally, recent advances in computational hardware create opportunities for addressing different parts of the problem more efficiently. Discrete subproblems can benefit from potentially quantum Ising solvers, while simulators and nonlinear solvers offer powerful tools for handling nonlinearities [33]. To fully exploit these emerging computational capabilities, we propose an integrated approach that decomposes MINLP problems into discrete and continuous components and solves each subproblem using the most suitable computational method [36]. In this work, two case studies are presented: an illustrative example involving the selection of an ionic liquid and its process design, and a more complex problem of drug substance manufacturing process optimization. The discrete subproblem in each case is formulated as an integer programming problem and solved using a commercial classical optimization solver, Gurobi. For comparative analysis, the same problem is reformulated as a quadratic unconstrained binary optimization and solved with simulated annealing, quantum annealing (QA), and entropy quantum computing (EQC). For the quantum methods, two different computing systems are used: D-Wave's specialized quantum annealer for QA, and Quantum Computing Incorporated (QCI)'s Dirac-1 quantum computer for EQC. The continuous subproblem is solved using Gurobi and a simulator-based optimization approach, respectively. In both examples, in terms of computational efficiency, Gurobi achieved the shortest runtime, whereas EQC took the longest, followed by QA and SA, in reaching feasible and optimal solutions. The heuristic methods demonstrated advantages in solution diversity compared to Gurobi's global search approach, identifying all or most of the feasible solutions in a single run and better capturing a broad solution space in a single run, while Gurobi provides global optimality guarantee and speed. This comparative analysis highlights the distinct

strengths of each method and underscores the potential of this heterogeneous computing approach, which enables the use of different methods to address practical optimization problems.

**Keywords:** process design · optimization · heterogeneous computing · quantum computing · quantum annealing

## 1   Introduction

Mixed-Integer Nonlinear Programming (MINLP) is an optimization framework that combines both discrete (integer or binary) and continuous decision variables, incorporating nonlinear relationships in the objective function and/or constraints, thereby enabling the modeling and solution of complex, real-world problems. Many applications, such as process design, operations research, and finance, involve decision-making problems that can be effectively modeled as MINLP problems [4,24]. One of the main challenges is that MINLP problems are complex to solve. In fact, the complexity of many practical problems are non-deterministic polynomial-time (NP) hard, and finding a good or even feasible solution can be challenging [28]. Various practical methods have been derived and solvers have advanced to address this complexity, but the problems and their scales that can be solved are still limited and smaller than what can be modeled [24]. One of the most common approaches to solving MINLP problems is to decompose the problem into smaller subproblems, commonly composed of discrete and continuous parts. The discrete subproblem can be solved to find candidate values for the integer variables, which can then be fixed, and the remaining problem can be solved as a nonlinear program (NLP) or passed to a simulator for evaluation. This approach allows for the use of specialized solvers that are designed to handle discrete variables and nonlinearities separately, thus making the overall problem more tractable. However, solving combinatorial optimization problems remains a challenge, as many of them are NP hard problems [12].

Ising solvers are being discussed as a potential candidate to address these problems, as these combinatorial problems can be modeled as Ising models and be solved with polynomial overhead [30]. Ising models, first developed to describe magnetism, can be used to describe the energy of physical systems through a Hamiltonian function:

$$H = \sum_{i \in V(G)} h_i \sigma_i + \sum_{(ij) \in E(G)} J_{ij} \sigma_i \sigma_j \tag{1}$$

where $\sigma_i \in \{-1, +1\}$ are binary spin variables, indexed by the vertices $V(G)$ of graph $G$, and the pairwise interactions are defined by the edges $E(G)$ of the graph, where $h_i$ and $J_{ij}$ are the corresponding coefficients [5,6]. Ising solvers are specialized hardware that minimize this energy function to find or approximate the ground states of the system [33].

Quadratic Unconstrained Binary Optimization (QUBO) is a mathematical formulation that can represent a wide range of combinatorial optimization problems. The Ising model can easily be mapped into a quadratic unconstrained

binary optimization (QUBO) and vice versa with a linear transformation of the spin variables $\sigma$ into binary variables $x$.

$$\min_{x} \quad \mathbf{x}^T \mathbf{Q} \mathbf{x} \quad \text{s.t.} \quad x \in \{0,1\}^n \tag{2}$$

where $\mathbf{Q}$ is an $n$-by-$n$ square, symmetric matrix of coefficients, and $\mathbf{x}$ is a set of binary variables. The Ising Hamiltonian can be expressed in the QUBO form with a simple change of variables $\sigma_i = 2x_i - 1$. QUBOs are particularly useful because many constrained integer programming (IP) problems can be reformulated into QUBOs. This is done by incorporating the constraints into the objective function as quadratic penalty terms to penalize infeasibility rather than imposing the constraints directly. This reformulation allows for the use of Ising solvers that are designed to solve Ising or QUBO problems [30,42].

A branch-and-bound (B&B) method is a classical deterministic approach to solving discrete optimization problems, which systematically explores the solution space by following a search tree and eliminating infeasible branches. Although B&B solver is not specifically designed for Ising or QUBO problems, it can also be used to solve Ising problems to global optimality [39]. In contrast to the classical solvers that strive to find the global optimum deterministically, Ising solvers are heuristic and probabilistic in nature. They report success probability, or likelihood of finding the optimal solution in a single run, without the global optimality guarantee Simulated annealing (SA) is a classical method that has traditionally been used to solve Ising or QUBO problems. In recent years, quantum methods such as quantum annealing (QA) and entropy quantum computing (EQC) have emerged as promising alternatives. The annealing techniques emulate the annealing process of metal thermal processing to attain the lowest lattice energy state; the analogy is that the optimal solution in a minimization optimization is found through a specific algorithmic treatment [22]. Simulated annealing (SA) uses random sampling to search the solution space for the optimal solution. It makes probabilistic decisions with the aim of escaping local minima and finding the global minimum energy [19]. Quantum annealing (QA) leverages quantum effects and quantum adiabatic evolution [16] to aim at finding the ground state of an optimization problem encoded in a Hamiltonian function [38]. Another quantum method is entropy quantum computing (EQC), which exploits the inherent noise and loss of quantum systems to promote the evolution of the lower-energy states of the Hamiltonian while suppressing the higher-energy states [37]. Emergence of these solvers and other alternative computing hardware offers new opportunities to tackle combinatorial problems, and recent advances in these hardware have made enough progress to warrant exploration of their potential in practical applications.

The main discussion in this work pertains to process superstructure optimization, yet the proposed framework and approach are not limited to this domain and generalizable to other mixed-integer programming (MIP) problems with decomposable structure. In process design and optimization, Ising solvers can be particularly useful for addressing the combinatorial aspects of design problems, such as equipment selection, process configuration, and operational scheduling.

A process synthesis optimization problem can be written as a MIP with an objective function subject to equality and inequality constraints, as shown below [32,41].

$$\min_{x,y} f(x,y) \quad \text{s.t.} \quad h(x,y) = 0, \quad g(x,y) \leq 0, \quad x \in \mathbb{R}^n \subseteq X, \quad y \in \mathbb{Z}^m \subseteq Y \quad (3)$$

where $x$ represents continuous variables (e.g., flow rate), $y$ represents integer variables (e.g., unit selection). $f$ represents the objective function, which is the performance metric of the optimization problem (e.g., cost). The model equations $h$ describe the interaction of state variables with system physics (e.g., mass balances), and the inequalities $g$ describe specifications and operational or safety constraints (e.g., critical quality attribute requirements). In the case of maximization, the negative value of the corresponding subject of interest can be minimized (e.g., the negative value of the production mass). The set of constraints define the feasible space of the problem, which is represented algebraically by $h$ and $g$. In cases where derivation of such algebraic expressions is difficult, simulation tools can be used to represent the process in a black-box optimization approach [9].

Using a MINLP for end-to-end optimization (E2EO) has been actively explored in relatively well-behaved system applications such as steady-state processes observed in the chemical industry [8,34,13]. However, in systems that require high modeling fidelity and exhibit complex behaviors, such as pharmaceutical processes, the mathematical programming approach to optimization has been limited in its applicability. The derivation of equation-oriented optimization models for these systems is complex, and simulation-based optimization requires an iterative process that can result in a non-trivial computational burden [3]. However, efforts have been made to integrate process synthesis optimization with overall process dynamics optimization. In Reference [3], the authors propose a rule-based and optimization-driven decision framework for optimizing flowsheets in a DS manufacturing process. The methodology leverages heuristic rules, such as regulatory considerations and knowledge-based rules, as well as scenario analysis, to generate a smaller search space. This framework efficiently narrows down the search space, thus evaluating alternative configurations more effectively overall. However, the derivation of these rules is susceptible to user bias and lacks a quantitative evaluation of the alternative configurations. In other words, this framework does not directly optimize the configuration selection itself.

To address the complexity of discreteness and desire for high fidelity in process design problems, we explore a heterogeneous computing approach that leverages the advantages of Ising form and use of involved optimization algorithms like black-box optimization. Though decomposition, we can isolate the discrete part of the problem to the Ising part of the formulation, offloading a source of hardness and better capturing the combinatorial part of the problem. Furthermore, we can still maintain high fidelity and solve the nonlinear part of the problem using black-box optimization approach. In this work, we investigate this approach and the application of the Ising solvers through both classical and

quantum algorithms to solve combinatorial problems, with a focus on process synthesis and practical decision-making.

## 1.1 Contributions of this work

The contributions of this work are as follows:

- propose a framework for integrating Ising solvers to solution algorithm of complex MINLP problems,
- application case studies of using Ising solvers, such as simulated annealing (SA), quantum annealing (QA), and entropy quantum computing (EQC), to solve discrete subproblems in process design problems,
- exploration of the use of quantum computing methods, such as quantum annealing (QA) and entropy quantum computing (EQC), in process design problems,
- provide open-source code for the case studies to facilitate reproducibility and further research in this area[1].

## 2 Methods

This section describes the methods used, including the formulation of discrete subproblems and the solution approaches employed. The continuous subproblems are not discussed in detail, as they are not the focus of this work, but they are solved using a black-box optimization approach with a simulator or an optimization solver. Two case studies are explored in this work: an illustrative example of ionic liquid selection and process configuration and a more complex drug substance manufacturing process optimization problem. B&B, SA, QA, and EQC are used to solve the discrete subproblem in both case studies.

## 2.1 Discrete Subproblem Formulation and Solution Methods

As many discrete variables in decision-making MINLP problems are binary (e.g., unit selection), the discrete subproblem is formulated as an integer programming (IP) problem with binary variables in this work. The formulation is implemented using `JuMP`[29] in `Julia` or using `Pyomo`[7] in `Python` programming language and solved with `Gurobi`[18]. To find all feasible solutions, an iterative process is implemented by adding a "no-good cut" to eliminate the current solution. In each iteration $n$, a "no-good cut" constraint is added to eliminate previous solution as infeasible as $\sum_{y|y_{n-1}=1}(1-y) + \sum_{y|y_{n-1}=0} y \geq 1$ for $n = \{1, 2, \ldots, n_{max} - 1\}$ with $n_{max} = N$ or the number of possible alternatives. Each iteration in this approach represents solving of the combinatorial part of the MINLP, fixing of the discrete variables, and solving the resulting NLP subproblem for evaluation.

For other Ising solvers, the IP subproblem is reformulated into a QUBO problem through an open-source `Julia` package, `QUBO.jl` [42], and solved using

---

[1] https://github.com/SECQUOIA/pd_ising

simulated annealing, quantum annealing, and entropy quantum computing. The simulated annealing (SA) method was implemented using the `dwave-neal` package [15] with default parameters, consisting of 1000 reads and 1000 sweeps. For quantum annealing (QA), the D-Wave Advantage 4.1 quantum processing unit (QPU) [14] was used with the default annealing schedule and a sample anneal time of 20 $\mu$s, with 1000 reads [6]. For entropy quantum computing (EQC), the QCI Dirac-1 device was used with 100 samples per run, which was the maximum number allowed [25,26].

For comparison in computation time, a performance metric called time-to-target (TTT) is introduced.

$$\text{TTT}_s = \tau \cdot \frac{\log(1 - s)}{\log(1 - p_{\text{target}})} \tag{4}$$

where $\text{TTT}_s$ is the time required to achieve success with probability $s$ (typically 0.99) in reaching the *target*. $\tau$ is the execution time of the algorithm. $p_{\text{target}}$ is the probability of reaching the target solution. If $p_{\text{target}} = 1$, which applies for deterministic solvers, then $\text{TTT}_s = \tau$ [31]. Time to *optimality* and finding *all feasible solutions* are considered as targets in this work. Targeting optimality serves as a benchmark for the performance of the methods, while targeting all feasible solutions provides a measure of the solution diversity and completeness of the search space. From a decision-making perspective, finding all feasible solutions is relevant, if not crucial, as it allows for a comprehensive understanding of the alternatives and enables better-informed decisions.

The computational setup used in this work runs on a Linux Ubuntu 22.04 operating system and features an Intel® i7-1365U processor with a base frequency of 1.80 GHz and 32.0 GB of RAM. The environment supports both Python 3.10.12 and Julia 1.11. Quantum computing resources include the D-Wave Advantage 4.1 quantum annealer and the QCI Dirac-1 entropy-based quantum computer. Optimization and modeling tasks were performed using the following packages: `JuMP` v1.26.0, `Pyomo` v6.7.3, `ToQUBO.jl` v0.1.10, `PharmaPy` v0.4.0, and `pyNOMAD` v4.4.0. Solver and hardware interfaces include `Gurobi` (v11.0, v12.0.2), `dwave-neal` v0.6.0, and `qci_client` v4.5.0.

## 2.2   Quadratic Unconstrained Binary Optimization (QUBO) Reformulation

A constrained IP problem with binary variables $y$ can be represented as Eq. (5); inequality constraints (e.g., $g$ in Eq. (3)) can be written into $Ay = b$ form by adding slack variables to the expressions [17].

$$\min_{y} \quad \mathbf{c}^T \mathbf{y} \quad \text{s.t.} \quad \mathbf{A}\mathbf{y} = \mathbf{b} \quad \mathbf{y} \in \{0, 1\}^n \tag{5}$$

where $\mathbf{c}$ is the cost vector, $\mathbf{A}$ is the constraint matrix, and $\mathbf{b}$ is the right-hand side vector.

These problems can be reformulated into a QUBO model by including quadratic infeasibility penalties in the objective function. The constraints $Ay = b$ are moved into the objective function with a penalty cost ($\rho$) as:

$$\min_{y} \quad \mathbf{c}^T\mathbf{y} + \rho(\mathbf{Ay} - \mathbf{b})^T(\mathbf{Ay} - \mathbf{b}) \tag{6}$$

The penalty terms are: $\rho(\mathbf{Ay} - \mathbf{b})^T(\mathbf{Ay} - \mathbf{b}) = \rho(\mathbf{y^T}(\mathbf{A^T A})\mathbf{y} - 2(\mathbf{A^T b})\mathbf{y} + \mathbf{b^T b})$. Taking advantage of $y^2 = y$ for $y \in \{0, 1\}$, the linear terms ($c$, $A^T b$) appear on the diagonal of the matrix $Q$ [17].

The general form of a QUBO problem can be written as described in the following:

$$\min_{z} \quad \mathbf{z}^T\mathbf{Qz} \quad \text{s.t.} \quad \mathbf{z} \in \{0, 1\}^n \tag{7}$$

where $\mathbf{z}$ is a set of binary variables; $\mathbf{Q}$ matrix is an $n$-by-$n$ square, symmetric matrix of coefficients [23].

### 2.3   Illustrative example: discrete subproblem formulation

The original problem is (P8) in Reference [20]; its formulation is included in the Supplementary section for completeness. This case study presents an optimization problem involving the synthesis of a reactor-separator network with two reactors and three separator options while simultaneously selecting an ionic pair from a list of two cations and two anions for the process. There are a total of 84 possible combinations of the discrete choices. When formulated as a single monolithic problem, it is a MINLP with a nonlinear objective function and binary and continuous variables. This problem was decomposed into two subproblems: 1) the discrete network synthesis and ion pair selection problem, and 2) the continuous flow optimization at fixed discrete variables. The discrete subproblem was formulated as an integer program (IP) with the objective function to minimize the total cost and binary variables for cation ($z_c$) and anion ($z_a$) selection, flow ($f_{i,j}$) and unit selection ($y_{r/s}$) as follows:

$$\min_{y,w} \quad \sum_{k \in \mathcal{K}} c_k^{\text{fixed}} y_k + 2 \sum_{r \in \mathcal{R}} c_r^{\text{oper}} y_r \alpha_r + 2 \sum_{s \in \mathcal{S}} \sum_{c \in C} \sum_{a \in A} c_s^{\text{oper}} y_s \beta_{s,c,a} w_{c,a} \tag{8}$$

$$\text{s.t.} \ f_{\text{src},r} = y_r \qquad \forall r \in \mathcal{R} \tag{9}$$

$$f_{s,\text{sink}} = y_s \qquad \forall s \in \mathcal{S} \tag{10}$$

$$\sum_{r \in \mathcal{R}} f_{\text{src},r} - f_{\text{src},r_1} \cdot f_{\text{src},r_2} = 1 \tag{11}$$

$$\sum_{s \in \mathcal{S}} f_{s,\text{sink}} \geq 1 \tag{12}$$

$$f_{r,s} \cdot f_{\text{src},r} = f_{r,s}, \quad f_{r,s} \cdot f_{s,\text{sink}} = f_{r,s} \qquad \forall r \in \mathcal{R}, s \in \mathcal{S} \tag{13}$$

$$(1 - f_{\text{src},r}) + \sum_{s \in \mathcal{S}} f_{r,s} \geq 1 \qquad \forall r \in \mathcal{R} \tag{14}$$

$$(1 - f_{s,\text{sink}}) + \sum_{r \in \mathcal{R}} f_{r,s} \geq 1 \qquad \forall s \in \mathcal{S} \tag{15}$$

$$\sum_{c \in C} z_c = 1, \quad \sum_{a \in A} z_a = 1 \tag{16}$$

$$w_{c,a} = z_c \cdot z_a \qquad \forall c \in C, a \in A \tag{17}$$

where $\mathcal{R}$ is the set of reactors, $\mathcal{S}$ is the set of separators, and $\mathcal{R}, \mathcal{S} \subset \mathcal{K}$ are the sets of units in the overall process. $C$ is the set of cations, and $A$ is the set of anions. Additionally, $c_k^{\text{fixed}}$ is the fixed cost of unit $k$, $c_{r/s}^{\text{oper}}$ is the operating cost of reactor or separator $r/s$, $\alpha_r$ is the conversion factor of reactor $r$, and $\beta_{s,c,a}$ is the separation factor of separator $s$ for cation $c$ and anion $a$. Constraints for the subproblem are formulated with domain knowledge and logic-based statements; for example, flow conservation such as  *if there is flow into a reactor, there must be flow into at least one of the separators* is enforced through a constraint, $(1 - f_{src,r}) + \sum f_{r,s} \geq 1$, as shown in Eq. (14). Eqs. (9) and (10) ensure that the flow from the source to the reactor and from the separator to the sink is equal to the binary variable of unit selection, $y_r$ and $y_s$, respectively. Constraints (11) and (12) ensure that there is at least one flow from the source to the reactor and from the separator to the sink, and constraints (13)-(15) ensure that the flow conservation is satisfied at each unit. Lastly, a cation and an anion are selected using Eq. (16), and the product of the two selections is represented by a binary variable $w_{c,a}$ in Eq. (17).

## 2.4 Simulation-based optimization problem example: discrete subproblem formulation

This case study presents a simulation-based optimization problem of a drug substance (DS) manufacturing process and serves as a case study of a more complex problem than the illustrative example. The original problem is adopted from [11] and describes a process of synthesis of a general drug substance, consisting of reaction, crystallization, and separation steps with multiple operating options for each step. The process configuration features two reactors, an evaporator for the solvent switch step in preparation for the crystallization step, and a filtration step to separate the solid active pharmaceutical ingredient (API). Three reactor types are considered for each reaction unit: a plug-flow reactor (PFR), and two continuously stirred tank reactors (CSTR) with different operating modes (continuous and batch). For evaporation, only the batch process is considered. Four options are considered for the crystallization step: one option is a batch crystallizer, and the other three options are continuous mixed suspension mixed product removal units (MSMPR) in series, ranging from one to three units. For each step, only one option is selected.

The problem is decomposed into two subproblems: a discrete configuration selection problem and a simulation-based operational optimization problem. A superstructure of the process is shown in Figure 1. In total, there are 36 alternative configurations.

The objective function of the configuration design subproblem is to minimize the total capital expense, and the objective function for the simulation-based subproblem is to maximize the production rate and crystal size. The simulation-based optimization framework has been adopted from the literature [2,3,27], and is not discussed in detail; the primary focus of this work is the discrete problem and the application of the heterogeneous computing framework.

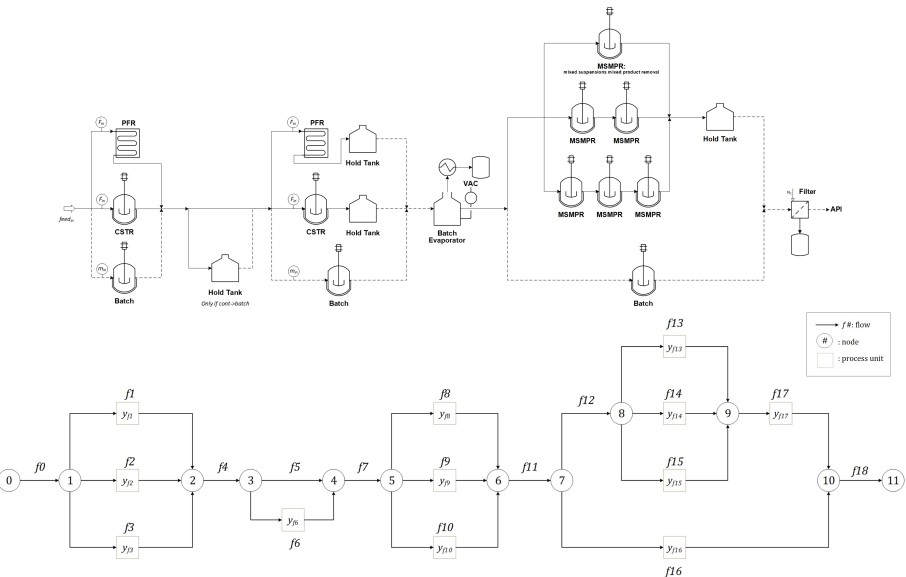

**Fig. 1.** A superstructure of a drug substance manufacturing process (top) and a representative flow diagram for the main problem (bottom); dotted and solid lines indicate batch and continuous feed, respectively.

Similarly to the illustrative example, the discrete problem is formulated as an integer program and as a QUBO problem, and solved using `Gurobi v.11.00` and the aforementioned heuristic methods (SA, QA, EQC) and the corresponding parameters, respectively. For the continuous subproblem, the configuration is fixed based on the solution of the discrete subproblem, and an open-package simulator, `PharmaPy`[10], is used to simulate the process and optimize the flow rates of the process, using `pyNOMAD`[1] as a black-box optimizer. The objective of the simulation-based framework is to maximize the production rate and crystal size.

Two sets of variables are defined for optimization of the process configuration: $F = \{f_{00}, f_{01}, ..., f_{18}\}$, representing the binary flow variables, and $L = \{y_{f_{01}}, y_{f_{02}}, ..., y_{f_{17}}\}$, representing the binary variables associated with the unit operation through which the corresponding flows pass, as illustrated in Figure 1. For a system with parallel units or flows at disjunctions, these two variables ($f$ and $y$) may need to be treated separately. In this case study, each disjunction represents a discrete choice of operating mode rather than a flow split; therefore, a single binary variable, $f$, is used to represent both flow and unit selection in this case. The objective function of the subproblem is to minimize the total capital expense, $C$. The unit capital cost ($c_i$) can be estimated in various ways, such as using previously calculated data. In cases where data are unavailable, theory-based calculations or approximations suffice; methods such as the bare

module method can be used to estimate the associated expense [40]. For flows that do not activate any unit, the associated cost is set to zero.

The formulation of the discrete subproblem is as follows:

$$\min_f \quad C = \sum_{i \in F} c_i f_i \tag{18}$$

$$\text{s.t.} \quad \sum_{j \text{ inflow to unit } l} f_j = \sum_{k \text{ outflow to unit } l} f_k \quad \forall\, l \text{ (for all units)} \tag{19}$$

$$g(f) \leq 0 \quad f_i \in \{0,1\} \quad \forall\, i \text{ (for all flows)} \tag{20}$$

Three types of constraints are declared: flow conservation at each node (19), single-unit selection at each disjunction, and logic-based constraints. At each disjunction, only one selection of unit or flow rule is enforced through an inequality constraint such as $(f_{01} + f_{02} + f_{03} \leq 1)$. Logic-based constraints, such as *when a continuous unit is followed by a batch process, a holding tank must be placed in between these two units*, are incorporated into the optimization framework as inequality constraints $(g(f) \leq 0)$. For example, $(1 - f_{01}) + (1 - f_{10}) + f_{06} \geq 1$, this equation indicates that if the first reactor is PFR ($f_{01} = 1$) and the second reactor is a batch reactor ($f_{10} = 1$), then the holding tank must be installed ($f_{06} = 1$). The full formulation of the discrete subproblem is included in the Supplementary section (Section **??**) for completeness.

## 3   Results

### 3.1   Illustrative example: Ionic Liquid Selection and Configuration of Reactor-Separator Network (IL)

The results of the discrete subproblems are presented in this section, and the detailed formulation and approach are described in Section 2.3.

The bar graph results presented in Figure 2 show the energy or probability of solutions found in a single run of the heuristic methods in the y-axis, ranked by the discrete problem's objective function value on the x-axis. The plot illustrates that out of 84 possible configurations identified by Gurobi, SA was able to find all 84 feasible solutions in a single run, while QA and EQC found 66 and 30 feasible solutions, respectively. All methods were able to find the optimal solution.

The algorithm execution time values along with the calculated time-to-target (TTT) metrics are summarized in Table 3.1. Overall, Gurobi achieved the fastest computation time for both finding the optimal solution and identifying all feasible solutions, followed by SA, QA, and EQC.

These results highlight the differences among these methods. The heuristic methods can explore all feasible solutions in a single run, but require longer times to do so. In contrast, IP quickly solves for optimality, albeit with multiple iterations corresponding to the problem size, to discover all feasible solutions.

The optimal discrete choices from the original MINLP were found in the 72nd iteration solution of Gurobi for the discrete subproblem. This occurred from the misalignment of the objective functions in the two problems. Although the objective function of the discrete subproblem was formulated to approximate that

of the original problem, it could not account for the complexities involved with the continuous variables in the original problem. As a result, the subproblem solutions did not replicate the exact ranking of solutions in the iterations. Nevertheless, the integer program formulation of the discrete subproblem allowed the use of the discrete solvers and other optimization techniques.

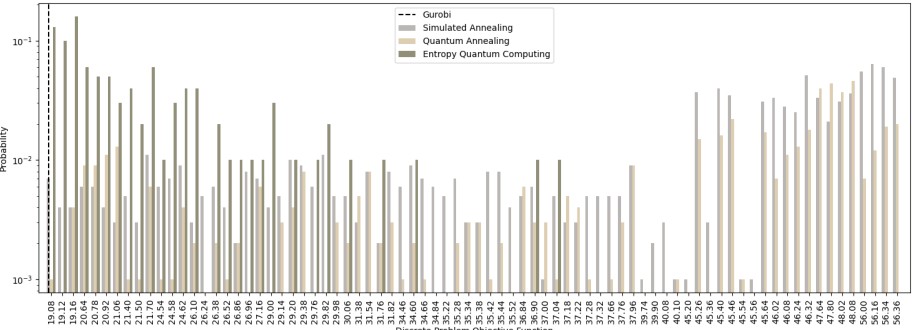

**Fig. 2.** A plot of energy or probability of solutions found through various methods in IL case study. The vertical line indicates the optimal solution of the discrete subproblem found by `Gurobi`. Infeasible solutions are not shown in this plot.

**Table 1.** Summary table of time to target (Opt: optimality, Feas: all feasible solutions) results of SA, QA (DWave Advantage), and EQC (QCI Dirac 1) for discrete subproblem of the illustrative problem. All times are in seconds. The subscript "quantum" indicates the time required for the quantum processing unit only, while "total" includes both communication overhead and the time for the quantum processing unit. The subscript "device" indicates the computing time inclusive of quantum and processing time[21].

| Solution Method | Execution Time | Time to Target ($TTT_{99}$) | |
| --- | --- | --- | --- |
| | $\tau$ | $TTOpt_{99}$ | $TTFeas_{99}$ |
| IP (Gurobi) | 0.003 | 0.003 $(= \tau)$ | 0.477 |
| SA | 0.34 | 221.7 | 0.53 |
| $QA_{quantum}$ | 0.13 | 620.5 | - |
| $QA_{total}$ | 1.59 | 7313.9 | - |
| $EQC_{device}$ | 37 | 1223.5 | - |

### 3.2   Simulation-based optimization problem example: Drug Substance Manufacturing Process Optimization

All results of the discrete subproblem are plotted in Figure 3; the simulation objective values at each iteration of IP are plotted along with the best objective value found with the iterations in Figure 4. The TTT metrics are presented in Table 3.2.

Similar trends were observed in this case study as the illustrative example. All methods were able to find the optimal solution. With a single run, SA identified all 36 feasible solutions, while QA and EQC found 28 and 6 feasible solutions, respectively. IP achieved the fastest computation time for both finding the optimal solution and identifying all feasible solutions, while entropy quantum computing exhibited the slowest computation time.

The best simulation objective value matched the 34th iteration in the IP method. The result of the discrete subproblem did not necessarily improve the value of the objective function of the simulations with each iteration, due to the misalignment of the objective functions in the two frameworks. Although multi-objective optimization can simultaneously optimize the overall system for several purposes (e.g., minimizing capital cost while maximizing product purity and production rate), the misalignment prevents efficient convergence of the two problems. An alignment of objectives between the two frameworks could accelerate the finding of the optimal solution.

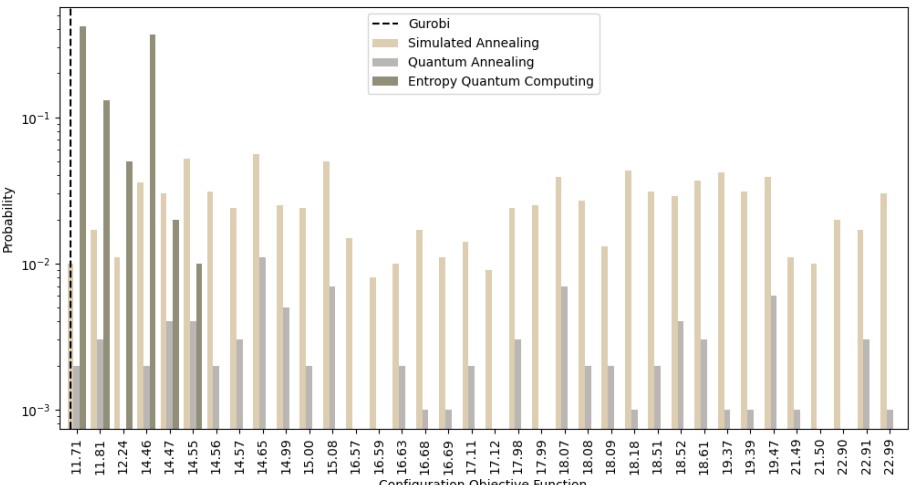

**Fig. 3.** A plot of energy or probability of feasible solutions found through annealing methods. The vertical line indicates the optimal solution of the discrete subproblem, indicated by the first result of `Gurobi`. Infeasible solutions are not shown in this plot.

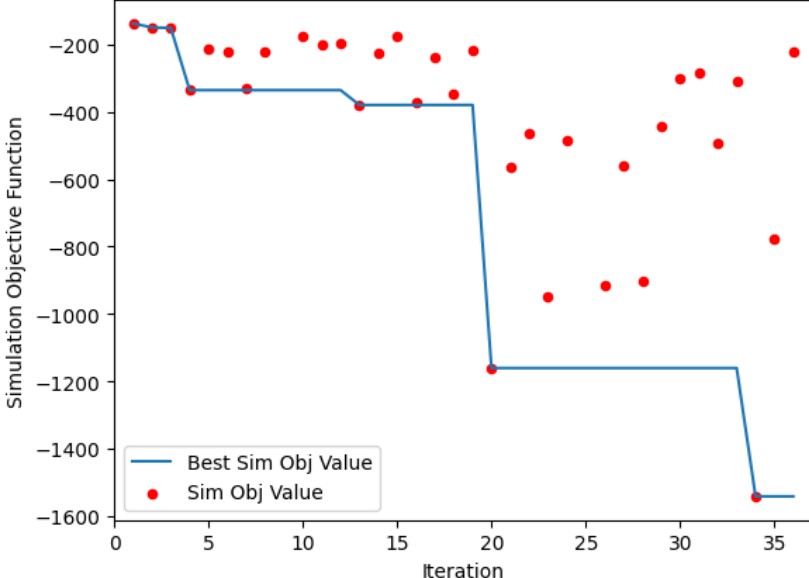

**Fig. 4.** Simulation objective function values ranked according to discrete subproblem solution by IP iterations (x-axis). Red dots indicate the objective function value of the corresponding simulation, and the blue line shows the best objective function value found with each iteration.

**Table 2.** Summary table of time to target (Opt: optimality, Feas: all feasible solutions) results of SA, QA (DWave Advantage), and EQC (QCI Dirac 1) for discrete subproblem of the drug substance manufacturing problem. All times are in seconds. The subscript "quantum" indicates the time for the quantum processing unit only, while "total" includes communication overhead. The subscript "device" indicates the computing time inclusive of quantum and processing time[21].

| Solution Method | Execution Time | Time to Target ($TTT_{99}$) | |
|---|---|---|---|
| | $\tau$ | $TTOpt_{99}$ | $TTFeas_{99}$ |
| IP (Gurobi) | 0.0009 | 0.0009 ($= \tau$) | 0.0790 |
| SA | 0.606 | 197.8 | 1.1 |
| $QA_{quantum}$ | 0.136 | 311.9 | - |
| $QA_{total}$ | 0.410 | 942.7 | - |
| $EQC_{device}$ | 35.0 | 295.9 | - |

## 4   Discussion

In general performance, the heuristic methods (SA, QA, EQC) are slower than `Gurobi` in finding the optimal solution, but they are able to find all feasible solutions in a single run, whereas `Gurobi` requires multiple iterations to find all feasible solutions. This effort grows exponentially with the size of the problem. In other words, heuristic methods can explore the solution space more broadly and provide diverse solutions within a single run, whereas the deterministic method, such as `Gurobi`, can offer a global optimality guarantee and speed. Still, it requires additional implementations, such as cuts, to thoroughly explore the solution space. Among the quantum methods, EQC showed the slower computation time and identified fewer solutions compared to QA. Although QA solutions were more broadly distributed across the solution space, EQC solutions were more clustered near the optimal solution. This indicates that EQC is able to find the best solutions first, but it may not explore the solution space as much as the other methods.

A clear understanding of these methodological differences can ensure that they are better leveraged in practical applications. The solution diversity of the heuristic methods may be particularly useful in realistic applications where the solution space is large and complex, and finding a feasible solution or alternatives is more critical than ensuring global optimality. For practical decision-making processes, the size of the optimization problem is often large, and exploring all combinatorial solutions is intractable and inefficient. In cases like this, heuristic methods can be employed to explore the solution space and identify feasible discrete solutions quickly. Several decisions can be made based on the heuristics and practical constraints such as cost, time, and resources, and then the remaining continuous problem can be solved to complete the optimization process. For instance, in a process design problem, the configuration decision can be made with the heuristics and impractical process choices can be eliminated in the first step, then the selected few configurations may proceed with simulations in parallel to fully understand the process implications/impact.

Choosing the proper computing method for the correct optimization problems can influence the computational effort required to solve these problems. In the IL case study, several degenerate solutions were observed in the subproblem, where multiple distinct feasible solutions resulted in the same objective function value. This illustrates one of the key differences between heuristic methods and deterministic solvers like `Gurobi`. In cases of degeneracy, deterministic solvers cannot prune any branches with the same objective value, and in the worst-case scenario, would have to evaluate all combinations of the discrete choices, which can increase exponentially with the problem size. Even worse, if cuts based on the objective value were imposed instead of an integer cut, there is a risk of eliminating alternative feasible, even optimal, solutions to the original problem. Additionally, for a simple problem such as the IL selection, current classical solvers, such as `Gurobi v.12.0`, can solve the original MINLP without modifications to the nonlinear objective function. Further processing of the problem into a QUBO problem actually introduces additional slack variables and con-

straints, unnecessarily increasing the problem size and complexity. In contrast, complex problems such as drug substance manufacturing process optimization, characterized by differential-algebraic equations (DAE), nonlinearities, and discrete decisions, are intractable when approached as a single monolithic MINLP. By applying a decomposition strategy, we can systematically integrate discrete configuration decisions into large-scale problems, such as these, and find practical solutions.

Lastly, it is essential to note that since these are different methods, different parameters were used, and their performance results may not be directly comparable. For heuristic methods, the parameters were set to their default values, and a limit of 100 samples was imposed for EQC. There is a potential for performance improvement through parameter tuning and by implementing an iterative process of adding cuts similar to that used in the IP method [35].

## 5   Conclusion

This work proposes a heterogeneous computing approach to solving MINLPs through a decomposition strategy with Ising solvers and other NLP solution methods. Various Ising solvers were explored and compared for computational performance and solution quality in one simple and one complex case studies of process design optimization. The integer programming (IP) formulation was first used to represent the discrete subproblem for B&B, and the QUBO reformulation was applied to express the discrete problem in a form suitable for Ising solvers.

The differences in the algorithm execution along with optimality guarantee and solution diversity of their results were discussed. IP method allowed us to find the global optimal solution to the subproblem, and it required iterations to find additional feasible solutions for practical decision-making. Other methods provided a probability distribution of solutions with a single execution of the algorithms, but not all methods were able to find the entire set of feasible solutions. Simulated annealing (SA) was able to find all feasible solutions while quantum methods (QA and EQC) did not find all feasible solutions in a single run. These results highlight that a better understanding of the differences in the solution methods can help in better leveraging these methods and hardware in practice.

**Acknowledgments.** This material is based upon work supported by the Center for Quantum Technologies under the Industry-University Cooperative Research Center Program at the US National Science Foundation under Grant No. 2224960.

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
