# OpenReview forum: "Leveraging Classical and Quantum Methods for Process Systems Engineering Applications"
_purdue.edu/Purdue_University/PQAI/2025/Symposium — PQAI 2025 Poster_

### Official Review · Reviewer_T7uE · 2025-07-23
**Leveraging Classical and Quantum Methods for Process Systems Engineering Applications**

**Rating:** 7
**Confidence:** 3

**Review:**

Recomendation: Poster presentation

Good points
Achieving both novelty and practicality: In contrast to conventional MINLP, this paper proposes a practical approach of separating the problem into "continuous + discrete" by utilizing the Ising solver and assigning appropriate calculation methods to each. The application of the latest quantum algorithms to process design using commercial quantum devices such as D-Wave and QCI Dirac-1 is an advanced example. Experimental verification has also been carried out in two different industrial applications (ionic liquid selection and pharmaceutical manufacturing). A GitHub repository is available to ensure code reproducibility in the public repository, and the open approach is also highly appreciated.

Requiring improvement:
It seems that the algorithm is not explored in depth enough. The theoretical explanation of quantum annealing and EQC is somewhat superficial, and the discussion of algorithm settings and constraints seems shallow. Integration of discrete and continuous parts: Currently, the solution is limited to the separation of discrete and continuous parts, and if a direction for true hybrid optimization that integrates these is shown, the contribution will be even greater.

---

### Official Review · Reviewer_GQ8z · 2025-07-24
**Review of "Leveraging Classical and Quantum Methods for Process Systems Engineering Applications"**

**Rating:** 6
**Confidence:** 3

**Review:**

The paper is very well written and focuses on MINLP, which are challenging problems. Some minor comments:
1) The link to the repository wasn't working when I followed it- maybe it isn't set to public yet.
2) Details on the problem sizes and how many times the problem was run (and even mean run-time with variance) would be useful to determine impact. Details on problem parameters are needed to reproduce the work, but these may be available in the GitHub.
3) The choice of penalty parameter $\rho$ can impact the solution quality in hybrid algorithms like QAOA. Do the results hold steady for different values of $\rho$, or does adjusting it change the output?
4) It appears that only one problem of each size was run. More runs on varying problem sizes would be interesting to show scalability of the approach- is there a regime where $n$ becomes large enough that the proposed methods can solve problems faster and more accurately than Gurobi can?
5) A comparison to hybrid approaches for MINLPs such as the below reference would be interesting.

Ajagekar, Akshay, Kumail Al Hamoud, and Fengqi You. "Hybrid classical-quantum optimization techniques for solving mixed-integer programming problems in production scheduling." IEEE Transactions on Quantum Engineering 3 (2022): 1-16.

---

### Decision · Program_Chairs · 2025-07-29

Accept (Poster)